# Incidence and risk factors of *C. trachomatis* and *N. gonorrhoeae* among young women from the Western Cape, South Africa: The EVRI study

**Vita W. Jongen**[1]*, **Maarten F. Schim van der Loeff**[1,2], **Matthys H. Botha**[3], **Staci L. Sudenga**[4], **Martha E. Abrahamsen**[5], **Anna R. Giuliano**[5]

1 Department of Infectious Diseases, Public Health Service Amsterdam, Amsterdam, The Netherlands,
2 Amsterdam UMC, Univ of Amsterdam, Internal Medicine, Amsterdam Infection and Immunity (AII), Amsterdam, The Netherlands, 3 Department of Obstetrics and Gynaecology, Stellenbosch University, Cape Town, South Africa, 4 Division of Epidemiology, Vanderbilt University Medical Center, Nashville, Tennessee, United States of America, 5 Center for Immunization and Infection Research in Cancer, Moffitt Cancer Center, Tampa, Florida, United States of America

* vjongen@ggd.amsterdam.nl

**Data Availability Statement:** Due to the sensitive nature of the data, data are only available upon reasonable request. All data relevant to this study

## Abstract

### Objective

Young women in South Africa are highly affected by sexually transmitted infections (STI), like *C. trachomatis* (CT) and *N. gonorrhoeae* (NG). We aimed to estimate the incidence of CT and NG, and its determinants, among young women from the Western Cape, South Africa, participating in an HPV vaccine trial (the EVRI study).

### Methods

HIV-negative women aged 16–24 years were enrolled between October 2012 and July 2013. At enrolment and month 6 participants were screened for CT and NG (Anyplex CT/NG real-time detection method). A questionnaire on demographic and sexual history characteristics was completed at enrolment and month 7. Treatment for CT and/or NG was offered to infected participants. Incidence rates (IR) of CT and NG were estimated. Determinants of incident CT and NG infections were assessed using Poisson regression.

### Results

365 women were tested for CT and/or NG at least twice. Prevalence of CT and NG at baseline was 33.7% and 10.4%, respectively. Prevalence of co-infection with CT and NG was 7.1%. During 113.3 person-years (py), 48 incident CT infections were diagnosed (IR = 42.4 per 100 py, 95% confidence interval (CI) 31.9–56.2). Twenty-nine incident NG were diagnosed during 139.3 py (IR = 20.8 per 100 py, 95%CI 14.5–29.9). Prevalent CT infection at baseline was associated with incident CT (adjusted incidence rate ratio (aIRR) 5.8, 95%CI 3.0–11.23. More than three lifetime sex partners increased the risk for incident NG (3–4 partners aIRR = 7.3, 95%CI 2.1–26.0; ≥5 partners aIRR = 4.3, 95%CI 1.1–17.5).

are included in the article or uploaded as supplementary material. Study dataset with deidentified participant data and protocol available from Anna.Giuliano@moffitt.org. Data requests can also be send to the Center for Immunization and Infection Research, which is the institutional body that handles data requests. Their email is: CIIRC@moffitt.org.

**Funding:** Supported in part by a research grant from Investigator-Initiated Studies Program of Merck Sharp & Dohme Corp (IISP39582). The opinion expressed in this paper are those of the authors and do not necessarily represent those of Merck Sharp & Dohme Corp. The funder provided support in the form of salaries for authors [ARG], but did not have any additional role in the analysis, decision to publish, or preparation of the manuscript. The specific roles of these authors are articulated in the 'author contributions' section. Dr. Sudenga (K07 CA225404) was supported by the National Cancer Institute.

**Competing interests:** The authors have read the journal's policy and have the following competing interests: ARG received a grant (IISP39582) from Merck during the course of this study and is a member of Merck research advisory boards. This does not alter our adherence to PLOS ONE policies on sharing data and materials. ARG and SLS received research funding from Merck outside the scope of this study. MFSvdL received research funding from Sanofi-Pasteur MSD; he is a co-investigator in a Sanofi-Pasteur-MSD HPV vaccine trial; he sat on a vaccine advisory board of GSK; his institution received in-kind contribution for an HPV study from Stichting Pathologie Onderzoek en Ontwikkeling (SPOO); his institution receives research funding from Janssen Infectious Diseases and Vaccines. There are no patents, products in development or marketed products associated with this research to declare. This does not alter our adherence to PLOS ONE policies on sharing data and materials. The other authors declare no conflicts of interest.

## Conclusions

The IR of bacterial STIs among young women in the Western Cape is very high. Besides being previously infected and a higher lifetime number of sex partners, no other risk factors were found for CT and NG, suggesting that the majority of these women were at risk. This indicates the need for intensified prevention of STIs as well as screening and treatment programs to increase sexual health in this region.

## Introduction

*C. trachomatis* (CT) and *N. gonorrhoeae* (NG) are among the most common bacterial sexually transmitted infections (STIs) worldwide [1, 2]. It has been shown that the relationship between HIV and STIs is bidirectional, where each increases the risk of the other [3, 4]. South Africa has one of the highest burdens of HIV [5, 6] and other STIs, such as CT and NG worldwide [7, 8]. In South Africa, young women are disproportionally affected by these STIs in comparison to males in the same age group [5, 7, 9–11]. This increased risk may be due to a combination of factors like age disparity and gender inequity in (sexual) relationships, early age of sexual debut, and co-infection with other STIs [9, 12–15].

The majority of infections with CT or NG in women are asymptomatic [16–19]. Untreated infections can cause long-term complications, including pelvic inflammatory disease, infertility, ectopic pregnancy and chronic pelvic pain [20]. In South Africa, STIs are managed syndromically [21]. Therefore, men and women without symptoms may remain largely untreated and may contribute to further transmission. Given the long-term complications and the public health risk of CT and NG for young women, more data are needed on the incidence of these infections in order to inform STI prevention and intervention programs. Although data on the prevalence of CT and NG are abundant, longitudinal data for non-pregnant, young HIV-negative women in South Africa are limited [14, 22–26]. Therefore, we aimed to estimate the incidence of CT and NG, and its determinants, among non-pregnant, young HIV-negative women from the Western Cape, South Africa, participating in the Efficacy of HPV Vaccine to Reduce HIV Infection (EVRI) Trial [27].

## Methods

### Study design and participants

The EVRI Trial (NCT01489527) enrolled young women living in the Western Cape, South Africa between October 2012 and July 2013. Full study procedures have been published elsewhere [27]. In brief, non-pregnant, HIV-negative women aged 16–24 years were randomized in a phase II randomized controlled trial to receive Gardasil (4-valent HPV vaccine) or a placebo. Participants received vaccination at enrolment, month 2, and month 6. Participants were followed an additional month after the final study agent dose was given, after which women from the placebo arm were offered the Gardasil vaccine.

This study was approved by the Institutional Review Boards of The University of South Florida, USA (number Pro00005120) and Stellenbosch University, South Africa (number N11/06/174). In addition, South African policies and ethical guidelines concerning parental permission for minors to participate in research studies were followed.

## Study procedures

Participants were screened for HIV at enrolment, month 2, month 6, and month 7. Urine samples for CT and NG testing were collected at enrolment and month 6. Reusable urine cups were used for the first 102 participants. Urine cups were thoroughly cleaned after every use using cidex immersion for at least 15 minutes, after which cups were rinsed with tap water. After a very high CT and NG prevalence was found at baseline [27] additional urine samples were collected in single use, disposable cups at month 2 (during which those diagnosed with CT and/or NG picked up of treatment for their infection) for the first 102 participants in order to rule out cross contamination. Afterwards, disposable cups were used. Participants who tested positive for HIV, CT or NG during follow-up were contacted, provided with STI results and offered to collect treatment at their earliest convenience. Treatment was dispensed by the local government clinic where also the study office was located. Acceptance and compliance to treatment was not captured. Sexual partners of the participant were not tested for STIs or notified of exposure to STIs during the course of this study. We considered a CT infection was adequately treated if a participant was provided with doxycycline, erythromycin, or a combination of metronidazole and doxycycline. We assumed an NG infection was adequately treated if a participant was provided with cefixime, ceftriaxone, or a combination of metronidazole, doxycycline and cefixime. Per the South African guidelines, counselling and free condoms were offered to all participants at every study visit.

At enrolment and the 7-month visit, participants completed a tablet-based questionnaire on sexual history, and socio-demographic characteristics (S1–S6 Files). To minimize recall and social desirability bias, the questionnaire was a computer-assisted self-interview (instead of interviewer-administered) and was available in English, Xhosa, and Afrikaans.

## Laboratory analyses

HIV status was assessed by rapid testing using Determine HIV-1/2 Ag/Ab Combo immunoassay (Alere Healthcare Pty Ltd, Waltham, MA, USA). HIV-seroconversion was confirmed using the Abbott AxSYM HIV Ag/Ab Combo (HIV Combo; Abbott, Wiesbaden, Germany) and the BioMerieux VIDAS HIV DUO assays (BioMeriux Inc., Durham, NC, USA). CT and NG were detected in urine specimens using the Anyplex CT/NG real-time detection method (Seegene, Seoul, South Korea). All laboratory analyses were performed in a central laboratory at the Tygerberg hospital (Cape Town, South Africa).

## Statistical analysis

CT, NG, and CT and/or NG prevalence at baseline was estimated by dividing the number of infections by the total number of participants. In order to rule out possible cross-contamination due to the use of re-usable cups (although thoroughly cleaned), we compared the prevalence at baseline and month 2 for the first 102 participants using Pearson's chi square test. If no evidence for cross-contamination was found, all these participants were included from their first study visit. If there was evidence for cross-contamination, all these participants would only be included from their second study visit (when the disposable cups were used).

Participants who were lost to follow-up after enrolment were excluded from the incidence analyses. Incidence rate of HIV per 100 person-years was estimated by dividing the number of incident infections by the total amount of person-years observation. We assumed that an incident HIV infection occurred at the midpoint between the last negative HIV test and the first positive diagnosis.

Participants with a prevalent CT or NG infection, who were not treated before their next study visit, were excluded from the respective incidence analyses. Participants with a prevalent

CT and NG infection at enrolment, who were not treated for one of both infections before their next visit, were excluded from the combined CT and/or NG incidence analysis. When participants received adequate treatment for CT and/or NG, we assumed they became at risk again for that infection 2 weeks after start of treatment. We defined an incident infection as the first positive CT and/or NG diagnosis after a negative CT and/or NG diagnoses or collection of adequate treatment. Incidence rates per 100 person-years of CT, NG, and CT and/or NG were estimated by dividing the number of incident infections by the total amount of person-years of observation.

Incidence rate ratios (IRR) and 95% confidence intervals (CI) of determinants of incident CT, NG and CT and/or NG infections were assessed using a Poisson regression model, using the number of incident infections as the outcome. Determinants associated in univariable analysis (at $p < 0.20$, Wald test) were included in the multivariable Poisson model. Age at enrolment, age at first sex act, and lifetime number of sex partners were forced into the multivariable model. Backwards selection, using the likelihood-ratio test, was performed to obtain the multivariable model with the best fit.

In a sensitivity analysis, we assessed the incidence rates per 100 person-years of CT, NG, and CT and/or NG, and the determinants of these incident infections, among participants without a prevalent CT, NG, or CT and NG infection, respectively.

All analyses were performed using Stata (version 15.1, StataCorp, College Station, TX, USA).

## Results

Of the 402 women who were enrolled and randomized in the EVRI trial, 381 had at least two study visits. Three out of 381 women were diagnosed with HIV during follow-up (incidence rate 1.4 per 100 person-years, 95% CI 0.5–4.5), of whom one was co-infected with both CT and NG at baseline. Three hundred sixty-five women were tested for CT and/or NG at least twice (Fig 1). Median age at baseline was 20 years [interquartile range (IQR) 19–22] (Table 1). The majority of women included at baseline reported that they had received education up to grade 12 or less (n = 212, 60.4%), were single (n = 334, 95.2%), and did not use birth control (n = 184, 52.4%). Median reported age of sexual debut was 17 years [IQR 16–18] and median reported number of lifetime sexual partners was 2 [IQR 1–3]. Thirteen women (3.7%) at baseline indicated that they had ever received presents, money or drugs in exchange for sex.

### Chlamydia

One hundred twenty-three women had a prevalent chlamydia infection at enrolment (prevalence 33.7%). Of the 173 women with a prevalent infection, 43 were not treated before their next study visit and were thus excluded from the incidence analysis (Fig 1). CT prevalence did not differ between enrolment (33.3%) and month 2 (29.2%) among women who were screened at both study visits (p = 0.59); therefore all visits of these women were thus included in the incidence analysis.

Among 322 women, 48 incident CT infections were diagnosed over 113.3 person-years (Table 2). The CT incidence rate was 42.4 per 100 person-years (95% CI 31.9–56.2). In multivariable analysis, having a CT infection at baseline was associated with an incident CT infection during follow-up (adjusted IRR (aIRR) 5.8, 95% CI 3.0–11.3) (Table 3, Fig 2). In sensitivity analysis among participants without a prevalent CT infection at baseline, the CT incidence rate was lower (29.0 per 100 person-years, 95% CI 20.0–42.0) and multivariable analysis did not identify determinants of an incident CT infection (S1 Table).

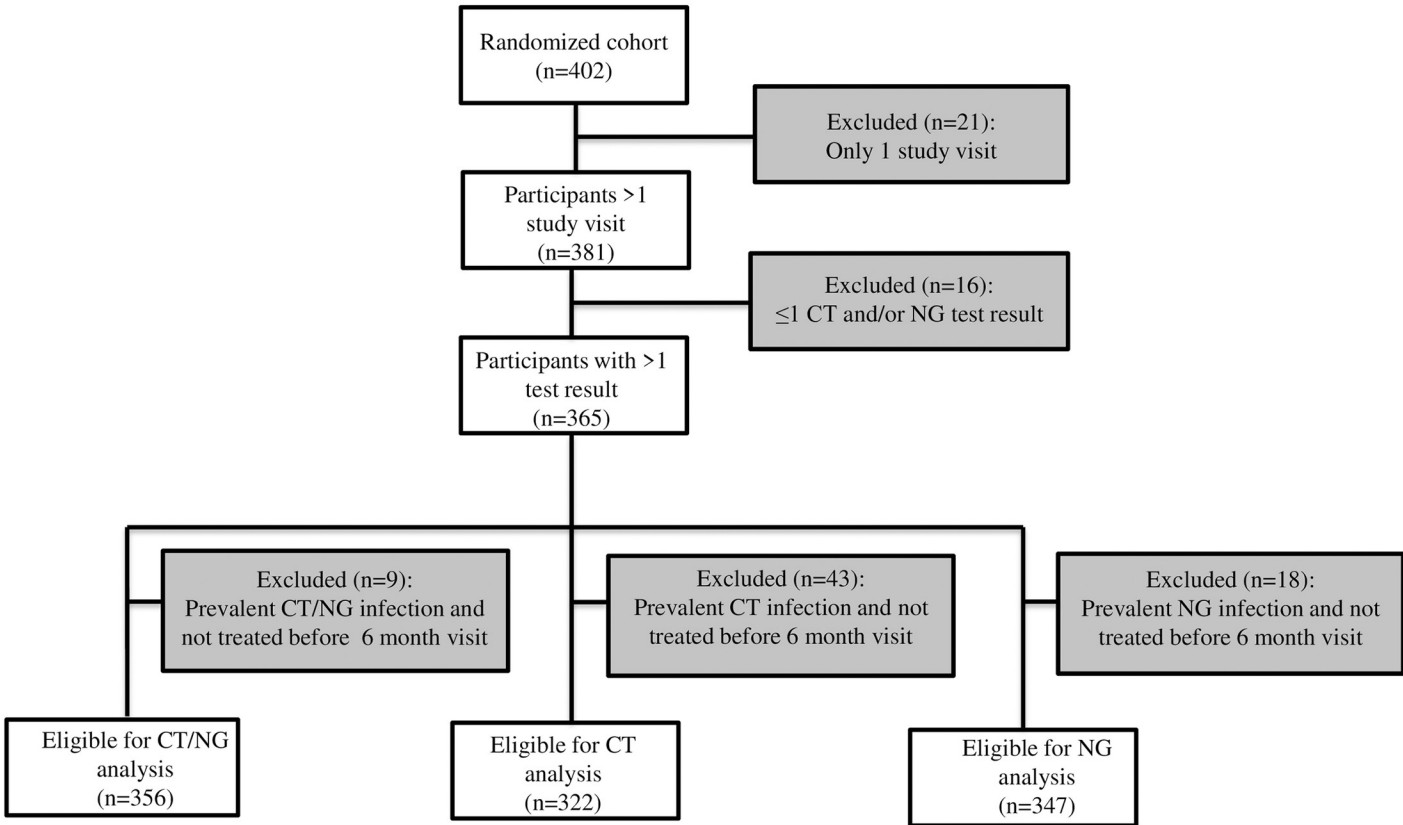

**Fig 1. Flowchart of participants who were excluded and included from the analysis on the incidence rate of chlamydia and gonorrhoea, EVRI study, Cape Town, South Africa, 2012–2014.**

## Gonorrhoea

Prevalence of NG at enrolment was 10.4% (38 among 365 women). Eighteen women with a prevalent NG infection were not treated before their next study visit and were therefore excluded from the incidence analysis. NG prevalence did not differ significantly between enrolment (8.3%) and month 2 (12.5%) among women who were screened for NG at both these study visits (p = 0.41); therefore all visits of these women were included in the incidence analysis.

Twenty-nine incident NG infections were diagnosed over 139.3 person-years, resulting in an incidence rate of 20.8 per 100 person-years (95% CI 14.5–29.9) (Table 2). In multivariable analysis, having three to four (aIRR 7.3, 95% CI 2.1–26.0) and five or more (aIRR 4.3, 95% CI 1.1–17.5) lifetime sex partners increased the risk of an incident NG infection (Table 3, Fig 2). In sensitivity analysis, NG incidence rate was 20.2 per 100 person-years (95% CI 13.9–29.5) and multivariable analysis yielded similar results (S1 Table).

## Chlamydia and/or gonorrhoea

Overall prevalence of CT and/or NG at enrolment was 37.0%. Twenty-six participants had both a prevalent CT and NG infection (prevalence 7.1%), of whom nine were treated for neither infection before their next study visit and were thus excluded from the incidence analysis (Fig 1).

**Table 1. Baseline socio-demographic and sexual behaviour characteristics of young women (n = 365) included in the EVRI study, Cape Town, South Africa, October 2012- July 2013.**

| | n[1] | %[1] |
|---|---|---|
| **Age (years)** | | |
| Median [IQR] | 20 | [19–22] |
| < 20 year | 141 | 38.6 |
| 20–21 year | 117 | 32.1 |
| ≥ 22 year | 107 | 29.3 |
| **Race[2]** | | |
| Black | 331 | 94.3 |
| Other | 20 | 5.7 |
| **Highest education level[2]** | | |
| ≤ Grade 12 | 212 | 60.4 |
| Passed grade 12 | 69 | 19.7 |
| Some college/tech | 70 | 19.9 |
| **Marital status[2]** | | |
| Single | 334 | 95.2 |
| Other | 17 | 4.8 |
| **Alcohol use in past month[2]** | | |
| No | 137 | 39.0 |
| Yes | 214 | 61.0 |
| **Current smoker[3]** | | |
| No | 286 | 96.3 |
| Yes | 11 | 3.7 |
| *Sex related characteristics* | | |
| **Age at first sex act (years)[4]** | | |
| Median [IQR] | 17 | [16–18] |
| < 16 year | 59 | 20.9 |
| 16–17 year | 140 | 49.7 |
| ≥ 18 year | 83 | 29.4 |
| **Ever been pregnant[2]** | | |
| No | 178 | 50.7 |
| Yes | 173 | 49.3 |
| **Current birth control use[2]** | | |
| No | 184 | 52.4 |
| Yes | 167 | 47.6 |
| **Type of birth control[5]** | | |
| Condom | 84 | 50.3 |
| Depo-Provera | 68 | 40.7 |
| Oral contraceptives | 10 | 6.0 |
| Other | 5 | 3.0 |
| **Lifetime number of sex partners[6]** | | |
| Median [IQR] | 2 | [1–3] |
| 0–1 | 130 | 37.4 |
| 2 | 75 | 21.6 |
| 3 | 74 | 21.3 |
| ≥ 4 | 69 | 19.8 |
| **Sex partners 6 months before baseline[2]** | | |
| Median [IQR] | 1 | [0–1] |

*(Continued)*

**Table 1.** (Continued)

| | n[1] | %[1] |
|---|---|---|
| 0 | 160 | 45.6 |
| 1 | 152 | 43.3 |
| ≥ 2 | 39 | 11.1 |
| **Condom use 6 months before baseline**[2,7] | | |
| Always | 80 | 41.9 |
| More than half of the time | 26 | 13.6 |
| Half of the time | 27 | 14.1 |
| Less than half of the time or never | 58 | 30.4 |
| **Ever received presents/money/drugs for sex**[2] | | |
| No | 338 | 96.3 |
| Yes | 13 | 3.7 |

**Abbreviations**: IQR, interquartile range;

[1]. Unless stated otherwise

[2]. Data missing for 14 participants

[3]. Data missing for 68 participants

[4]. Data missing for 83 participants

[5]. Among participants using birth control

[6]. Data missing for 17 participants

[7]. Among participants who had sex in the 6 months before baseline.

Among 356 women, 70 incident CT and/or NG infections were diagnosed over 148.8 person years. Overall incidence rate of CT and/or NG was 47.0 per 100 person-years (95% CI 37.2–59.5) (Table 2). In multivariable analysis, having a CT and/or NG infection at baseline (aIRR 2.5, 95% CI 1.5–4.4) and having 3–4 lifetime sex partners increased the risk of an incident CT and/or NG infection (aIRR 1.9, 95% CI 1.1–3.5) (Table 3, Fig 2). In sensitivity analysis, the incidence rate of CT and/or NG infections was lower (36.3 per 100-person-years, 95% CI 26.7–49.3) and having 3–4 lifetime sex partners was no longer significantly associated with an increased risk of an incident infection (S1 Table).

## Discussion

In this study we aimed to estimate the incidence rate of CT and NG among non-pregnant HIV-negative women living in the Western Cape, South Africa. We found a very high incidence rate of both CT and NG in this study population, even though participants were provided with counseling and free condoms during the trial. In addition, our data show that the young women in this study rapidly acquired one or both STIs after sexual debut. Having a prevalent CT infection at baseline and more than three lifetime sex partners were the only

**Table 2. Incidence rates of *C. trachomatis* and *N. gonorrhoeae* among young women in Cape Town, South Africa, October 2012-February 2014.**

| Infection | N at risk | Incident events | py | IR per 100 py | 95%CI |
|---|---|---|---|---|---|
| *C. trachomatis* | 322 | 48 | 113.3 | 42.4 | 31.9–56.2 |
| *N. gonorrhoeae* | 347 | 29 | 139.3 | 20.8 | 14.5–29.9 |
| *C. trachomatis* and/or *N. gonorrhoeae* | 356 | 70 | 148.8 | 47.0 | 37.2–59.5 |

**Abbreviations**: CI, confidence interval; IR, incidence rate; py, person-years.

**Table 3. Determinants of an incident chlamydia, gonorrhoea, and chlamydia and/or gonorrhoea infection, EVRI study, Cape Town, South Africa, October 2012-February 2014.**

| | Incident chlamydia (n = 48)[a] | | | | | | Incident gonorrhoea (n = 29)[b] | | | | | | Incident chlamydia and/or gonorrhoea (n = 70)[c] | | | | | |
|---|---|---|---|---|---|---|---|---|---|---|---|---|---|---|---|---|---|---|
| | IRR | (95% CI) | p | aIRR[d] | (95% CI) | p | IRR | (95% CI) | p | aIRR[e] | (95% CI) | p | IRR | (95% CI) | p | aIRR[f] | (95% CI) | p |
| **1. Socio-demographic characteristics** | | | | | | | | | | | | | | | | | | |
| Age, continuous[g] | 0.77 | (0.19;3.07) | 0.71 | 2.92 | (0.53;16.1) | 0.22 | 0.79 | (0.13;4.76) | 0.80 | 0.76 | (0.09;6.37) | 0.80 | 0.69 | (0.21;2.19) | 0.52 | 1.34 | (0.33;5.39) | 0.68 |
| **Age, categorical** | | | | | | | | | | | | | | | | | | |
| < 20 year | REF | | 0.76 | | | | REF | | 0.65 | | | | REF | | 0.66 | | | |
| 20–21 year | 0.97 | (0.50;1.89) | | | | | 1.24 | (0.54;2.87) | | | | | 0.92 | (0.53;1.59) | | | | |
| ≥ 22 year | 0.78 | (0.39;1.57) | | | | | 0.80 | (0.31;2.06) | | | | | 0.76 | (0.43;1.37) | | | | |
| **Race** | | | | | | | | | | | | | | | | | | |
| Black | REF | | 0.22 | | | | REF | | 0.75 | | | | REF | | 0.68 | | | |
| Other | 0.35 | (0.05;2.54) | | | | | 1.27 | (0.30;5.37) | | | | | 0.79 | (0.25;2.53) | | | | |
| **Highest education level** | | | | | | | | | | | | | | | | | | |
| ≤ Grade 12 | REF | | 0.77 | | | | REF | | 0.047 | | | | REF | | 0.10 | | | |
| Passed grade 12 | 0.86 | (0.40;1.86) | | | | | 0.37 | (0.11;1.27) | | | | | 0.63 | (0.32;1.24) | | | | |
| Some college/tech | 0.75 | (0.34;1.68) | | | | | 0.26 | (0.06;1.14) | | | | | 0.49 | (0.23;1.05) | | | | |
| **Marital status** | | | | | | | | | | | | | | | | | | |
| Single | REF | | 0.33 | | | | [h] | | | | | | REF | | 0.21 | | | |
| Other | 0.42 | (0.06;3.08) | | | | | | | | | | | 0.35 | (0.05;2.51) | | | | |
| **Alcohol use in past month** | | | | | | | | | | | | | | | | | | |
| No | REF | | 0.85 | | | | REF | | 0.06 | | | | REF | | 0.36 | | | |
| Yes | 0.94 | (0.50;1.76) | | | | | 2.45 | (0.90;6.63) | | | | | 1.28 | (0.74;2.21) | | | | |
| **Current smoker** | | | | | | | | | | | | | | | | | | |
| No | REF | | 0.03 | [i] | | | REF | | 0.31 | | | | REF | | 0.58 | | | |
| Yes | 0.18 | (0.03;1.34) | | | | | 1.82 | (0.62;5.37) | | | | | 0.78 | (0.31;1.95) | | | | |
| **2. Sex related characteristics** | | | | | | | | | | | | | | | | | | |
| Age at first sex act, continuous[g] | 1.87 | (0.24;14.46) | 0.55 | | | | 0.58 | (0.04;8.92) | 0.70 | | | | 1.00 | (0.17;5.76) | 1.00 | | | |
| **Age at first sex act, categorical** | | | | | | | | | | | | | | | | | | |
| < 16 year | REF | | 0.73 | REF | | | REF | | 0.32 | REF | | | REF | | 0.31 | REF | | |
| 16–17 year | 1.59 | (0.59;4.26) | | 1.17 | (0.42;3.24) | 0.77 | 1.90 | (0.54;6.68) | | 1.67 | (0.46;6.10) | 0.44 | 1.80 | (0.79;4.10) | | 1.57 | (0.67;3.68) | 0.29 |
| ≥ 18 year | 1.67 | (0.58;4.81) | | 1.00 | (0.32;3.10) | 1.00 | 0.95 | (0.21;4.24) | | 0.71 | (0.11;4.43) | 0.71 | 1.28 | (0.51;3.21) | | 1.05 | (0.39;2.85) | 0.92 |
| **Ever been pregnant** | | | | | | | | | | | | | | | | | | |
| No | REF | | 0.64 | | | | REF | | 0.35 | | | | REF | | 0.28 | | | |
| Yes | 1.15 | (0.64;2.08) | | | | | 1.42 | (0.67;3.01) | | | | | 1.31 | (0.80;2.14) | | | | |
| **Current birth control use** | | | | | | | | | | | | | | | | | | |
| No | REF | | 0.63 | | | | REF | | 0.18 | | | | REF | | 0.36 | | | |
| Yes | 1.17 | (0.61;2.27) | | | | | 1.90 | (0.70;5.16) | | | | | 1.30 | (0.74;2.29) | | | | |
| **Type of birth control** | | | | | | | | | | | | | | | | | | |
| No current use | REF | | 0.62 | | | | REF | | 0.07 | | | | REF | | 0.58 | | | |
| Condom | 1.40 | (0.69;2.83) | | | | | 1.30 | (0.41;4.10) | | | | | 1.30 | (0.70;2.42) | | | | |
| Depo-Provera | 0.99 | (0.40;2.49) | | | | | 2.18 | (0.66;7.13) | | | | | 1.17 | (0.55;2.50) | | | | |

*(Continued)*

**Table 3.** (Continued)

| | Incident chlamydia (n = 48)[a] | | | | | | Incident gonorrhoea (n = 29)[b] | | | | | | Incident chlamydia and/or gonorrhoea (n = 70)[c] | | | | | |
|---|---|---|---|---|---|---|---|---|---|---|---|---|---|---|---|---|---|---|
| | IRR | (95% CI) | p | aIRR[d] | (95% CI) | p | IRR | (95% CI) | p | aIRR[e] | (95% CI) | p | IRR | (95% CI) | p | aIRR[f] | (95% CI) | p |
| Oral contraceptives | 0.44 | (0.06;3.38) | | | | | 3.90 | (0.93;16.30) | | | | | 1.42 | (0.48;4.23) | | | | |
| Lifetime number of sex partners, continuous[j] | 1.12 | (0.67;1.84) | 0.67 | | | | 1.45 | (0.78;2.70) | 0.26 | | | | 1.24 | (0.82;1.87) | 0.31 | | | |
| **Lifetime number of sex partners, categorical** | | | | | | | | | | | | | | | | | | |
| < 3 | REF | | 0.35 | REF | | | REF | | 0.002 | REF | | 0.002 | REF | | 0.08 | REF | | |
| 3–4 | 1.52 | (0.77;3.01) | | 1.47 | (0.74;2.95) | 0.27 | 6.72 | (1.92;23.59) | | 7.29 | (2.05;26.00) | | 1.95 | (1.08;3.55) | | 1.93 | (1.06;3.53) | 0.03 |
| ≥ 5 | 0.86 | (0.36;2.08) | | 0.76 | (0.29;1.98) | 0.58 | 4.52 | (1.13;18.07) | | 4.32 | (1.07;17.46) | | 1.48 | (0.73;3.00) | 0.040 | 1.36 | (0.65;2.84) | 0.42 |
| **Sex partners since start study, continuous[j]** | 0.91 | (0.50;1.66) | 0.75 | | | | 0.95 | (0.40;2.22) | 0.90 | | | | 0.91 | (0.53;1.56) | 0.74 | | | |
| **Sex partners since start study, categorical** | | | | | | | | | | | | | | | | | | |
| 0 | REF | | 0.16 | | | | REF | | 0.87 | | | | REF | | 0.19 | | | |
| 1 | 2.50 | (0.76;8.25) | | | | | 1.41 | (0.32;6.31) | | | | | 2.19 | (0.78;6.15) | | | | |
| ≥ 2 | 1.59 | (0.44;5.69) | | | | | 1.48 | (0.31;6.98) | | | | | 1.56 | (0.52;4.67) | | | | |
| **Condom use since start study** | | | | | | | | | | | | | | | | | | |
| No sex | REF | | 0.57 | | | | REF | | 0.30 | | | | REF | | 0.33 | | | |
| Always | 1.98 | (0.59;6.64) | | | | | 1.11 | (0.24;5.12) | | | | | 1.77 | (0.62;5.06) | | | | |
| More than half of the time | 1.77 | (0.40;7.89) | | | | | 1.08 | (0.15;7.69) | | | | | 1.67 | (0.47;5.91) | | | | |
| Half of the time | 2.44 | (0.58;10.21) | | | | | 1.13 | (0.16;8.02) | | | | | 1.75 | (0.49;6.21) | | | | |
| Less than half of the time or never | 2.89 | (0.75;11.16) | | | | | 3.26 | (0.68;15.71) | | | | | 3.01 | (0.98;9.23) | | | | |
| **Sex for presents/money/drugs for sex since start study** | | | | | | | | | | | | | | | | | | |
| No | REF | | 0.74 | | | | REF | | 0.41 | | | | REF | | 0.47 | | | |
| Yes | 1.29 | (0.31;5.34) | | | | | 1.94 | (0.45;8.31) | | | | | 1.49 | (0.54;4.12) | | | | |
| **Prevalent infection at baseline[k]** | | | | | | | | | | | | | | | | | | |
| No | REF | | <0.001 | REF | | <0.001 | REF | | 0.52 | | | | REF | | 0.001 | REF | | 0.001 |
| Yes | 4.14 | (2.33–7.35) | <0.001 | 5.83 | (3.01;11.28) | <0.001 | 1.66 | (0.40;7.00) | 0.52 | | | | 2.23 | (1.38;3.58) | 0.001 | 2.54 | (1.47;4.38) | 0.001 |

**Abbreviations:** aIRR, adjusted incidence ratio; CI, confidence interval; IRR, incidence rate ratio; REF, reference category

[a]. 9 incident events were excluded from multivariable analysis due to missing values

[b]. 7 incident events were excluded from multivariable analysis due to missing values

[c]. 14 incident events were excluded from multivariable analysis due to missing values

[d]. Variables included in the multivariable model based on univariable analysis: age, lifetime number of sex partners, age at first sex act, prevalent chlamydia infection at baseline

[e]. Variables included in the multivariable model based on univariable analysis: age, lifetime number of sex partners, age at first sex act

[f]. Variables included in the multivariable mode based on univariable analysis: age, lifetime number of sex partners, age at first sex act, prevalent chlamydia and/or gonorrhoea infection at baseline

[g]. Per 10 year increase in age

[h]. Excluded from univariable and multivariable analysis due to 0 observations in one of the categories

[i]. Excluded from multivariable analysis due to being unstable in the model (large standard error and 95% confidence interval)

[j]. Per (log+1) increase in partner

[k]. Chlamydia, gonorrhoea, or chlamydia and/or gonorrhoea, respectively.

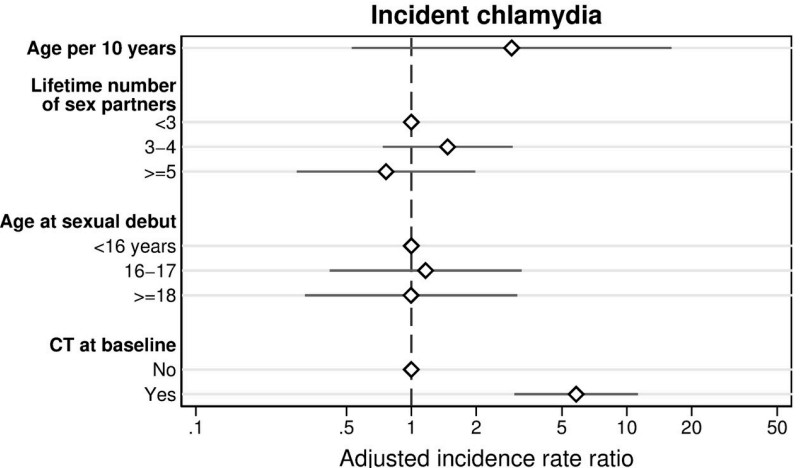

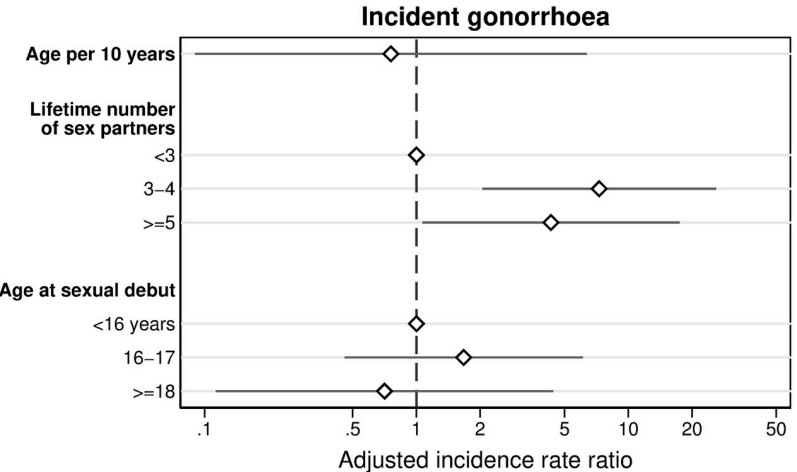

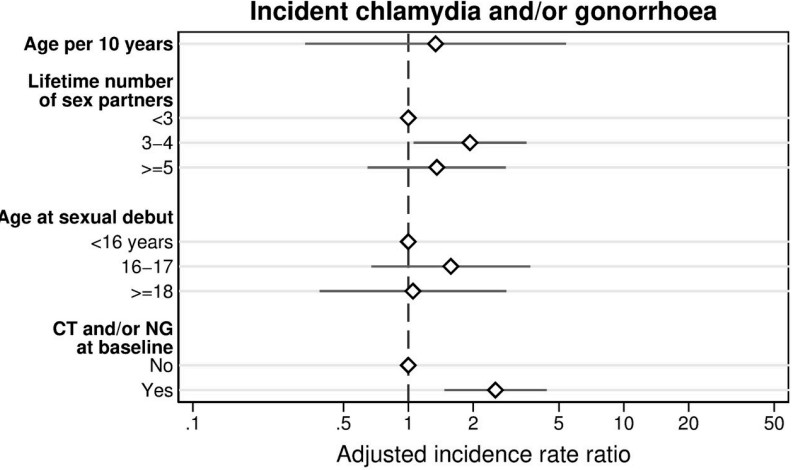

**Fig 2. Determinants of incident chlamydia, gonorrhoea, and chlamydia and/or gonorrhoea infections, result of multivariable Poisson regression analysis; EVRI study, Cape Town, South Africa, October 2012-February 2014.**

determinants associated with an increased risk of CT or NG, suggesting that the majority of these young women were at high risk for an incident infection.

The IR of both CT and NG in this study population is substantially higher than observed both within South Africa and across other countries in Sub-Saharan Africa [8, 22–26, 28]. Although only few lifetime sex partners (median of 2) were reported, STI incidence was high. In addition, a prevalent CT infection at baseline increased the risk 5-fold for an incident CT infection, similar to what was observed in another South African study [23]. As the STI prevalence in the general population of South Africa is high [7] and in the analysis we only included participants who received adequate treatment for their CT infection, it appears that repeated CT infections were common. Given that, even with a median of two lifetime sex partners, these women were at substantial risk for incident and repeated STI infections, suggesting the need to include both genders in STI screening programs in an effort to decrease STI risk. As the relationship between STIs and HIV is bidirectional [3, 4], intensified STI screening programs are essential for HIV control. In addition, annual screening and treatment of CT and NG could reduce pelvic inflammatory disease in women younger than 25 years of age [29].

Previous studies in South African populations found that younger women were at increased risk for incident CT or NG [14, 23, 26]. In addition, being single [14, 23], a higher number of sex partners [14], use of injectable contraceptives [14], alcohol consumption [23], and recent HIV acquisition [14, 26] increased the risk of CT and NG. As all women in our study were aged 25 years or younger, this might explain why we found no other determinants besides prevalent CT at baseline and number of lifetime sex partners. Due to the low number of incident HIV infections, HIV acquisition during follow-up could not be taken into account as a determinant.

A strength of this study is the unique study population of young women with a median age of 20 years. The following limitations should be taken into account when interpreting the results of this study. First, the results of this study may not be generalizable to all South African women and likely do not represent STI risk, and determinants, of women in other Sub-Sahara African countries. Second, although efforts were made to minimize recall and social desirability biases by using tablet based questionnaires in three languages, self-reported data on demographic characteristics and sexual history could still have been susceptible to these biases. Last, although adequate treatment was offered to participants with a CT or NG infection, no test of cure was performed. Therefore, it may be possible that some of the incident cases were the result of treatment failure or because the participant did not take the treatment. In sensitivity analyses among participants without prevalent infections at baseline, we showed that the incidence rate of CT, NG and CT and/or NG was still very high, although the CT incidence rate was somewhat lower than in the main analysis. The fact that the CT incidence rate was lower may be because participants with treatment failure or who did not take their treatment were now excluded, or because re-infection was more common among women with a previous CT infection.

## Conclusion

The IR of bacterial STIs among young women in the Western Cape is very high. Besides being previously infected with CT and a higher lifetime number of sex partners, no other risk factors were found for CT and NG, suggesting that the majority of these women were at high risk for both STIs. These data suggest that syndromic management of STIs may not be sufficient and indicate the need for intensified prevention of STIs. Moreover, given the high STI prevalence in the general population of South Africa, they indicate a need for gender neutral screening and treatment programs to increase sexual health in this region.

## Supporting information

**S1 File. Enrolment questionnaire (English).**
(PDF)

**S2 File. Enrolment questionnaire (Afrikaans).**
(PDF)

**S3 File. Enrolment questionnaire (Xhosa).**
(PDF)

**S4 File. Follow-up questionnaire (English).**
(PDF)

**S5 File. Follow-up questionnaire (Afrikaans).**
(PDF)

**S6 File. Follow-up questionnaire (Xhosa).**
(PDF)

**S1 Table. Sensitivity analysis of the determinants of an incident chlamydia, gonorrhoea, and chlamydia and/or gonorrhoea infection[a], EVRI study, Cape Town, South Africa, October 2012-February 2014.**
(DOCX)

## Acknowledgments

We would like to thank all the study participants, without whom this study would not have been possible. We further acknowledge the contributions of Charlotte Lawn, Wendy Adendorff, Zukiswa Gloria Ncume, Kayoko Kennedy, Dale Barrios, Jeannie Vaughn, David Jackson, Shahieda Isaacs, and Nafiisah Chotum.

## Author Contributions

**Conceptualization:** Staci L. Sudenga, Anna R. Giuliano.

**Data curation:** Martha E. Abrahamsen.

**Formal analysis:** Vita W. Jongen, Maarten F. Schim van der Loeff.

**Investigation:** Matthys H. Botha.

**Supervision:** Maarten F. Schim van der Loeff.

**Visualization:** Vita W. Jongen.

**Writing – original draft:** Vita W. Jongen.

**Writing – review & editing:** Maarten F. Schim van der Loeff, Matthys H. Botha, Staci L. Sudenga, Martha E. Abrahamsen, Anna R. Giuliano.

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
