## [Decision Letter · Decision Letter 0]

19 Nov 2020

PONE-D-20-31236

Incidence and risk factors of C. trachomatis and N. gonorrhoeae among young women from the Western Cape, South Africa: the EVRI study

PLOS ONE

Dear Dr. Jongen,

Thank you for submitting your manuscript to PLOS ONE. After careful consideration, we feel that it has merit but does not fully meet PLOS ONE’s publication criteria as it currently stands. Therefore, we invite you to submit a revised version of the manuscript that addresses the points raised during the review process.

Please pay specific attention to clarifying the methodological issues raised by the reviewers with regards to bias, definitions, and careful interpretation of the data in the absence of data of partner notification, ToC, etc.

We look forward to receiving your revised manuscript.

Kind regards,

Remco PH Peters, MD, PhD, DLSHTM

Academic Editor

PLOS ONE

Journal Requirements:

2. Please ensure you have included additional information regarding the survey or questionnaire used in the study and ensure that you have provided sufficient details that others could replicate the analyses. For instance, if you developed a questionnaire as part of this study and it is not under a copyright more restrictive than CC-BY, please include a copy, in both the original language and English, as Supporting Information, or include a citation if it has been published previously.

4.  Thank you for providing the following Funding Statement:  

 "Supported in part by a research grant from Investigator-Initiated Studies Program of Merck Sharp & Dohme Corp (IISP39582). The opinion expressed in this paper are those of the authors and do not necessarily represent those of Merck Sharp & Dohme Corp. Dr. Sudenga (K07 CA225404) was supported by the National Cancer Institute."

i) We note that one or more of the authors is affiliated with the funding organization, indicating the funder may have had some role in the design, data collection, analysis or preparation of your manuscript for publication; in other words, the funder played an indirect role through the participation of the co-authors.

If the funding organization did not play a role in the study design, data collection and analysis, decision to publish, or preparation of the manuscript and only provided financial support in the form of authors' salaries and/or research materials, please review your statements relating to the author contributions, and ensure you have specifically and accurately indicated the role(s) that these authors had in your study in the Author Contributions section of the online submission form. Please make any necessary amendments directly within this section of the online submission form.  Please also update your Funding Statement to include the following statement: “The funder provided support in the form of salaries for authors [insert relevant initials], but did not have any additional role in the study design, data collection and analysis, decision to publish, or preparation of the manuscript. The specific roles of these authors are articulated in the ‘author contributions’ section.”

If the funding organization did have an additional role, please state and explain that role within your Funding Statement.

ii) Please also provide an updated Competing Interests Statement declaring this commercial affiliation along with any other relevant declarations relating to employment, consultancy, patents, products in development, or marketed products, etc.  

iii) Within your Competing Interests Statement, please confirm that this commercial affiliation does not alter your adherence to all PLOS ONE policies on sharing data and materials by including the following statement: "This does not alter our adherence to  PLOS ONE policies on sharing data and materials.” (as detailed online in our guide for authors http://journals.plos.org/plosone/s/competing-interests). If this adherence statement is not accurate and  there are restrictions on sharing of data and/or materials, please state these. Please note that we cannot proceed with consideration of your article until this information has been declared.

Reviewers' comments:

Reviewer's Responses to Questions

**Comments to the Author**

1. Is the manuscript technically sound, and do the data support the conclusions?

Reviewer #1: Partly

Reviewer #2: Yes

2. Has the statistical analysis been performed appropriately and rigorously? 

Reviewer #1: Yes

Reviewer #2: Yes

3. Have the authors made all data underlying the findings in their manuscript fully available?

Reviewer #1: Yes

Reviewer #2: Yes

4. Is the manuscript presented in an intelligible fashion and written in standard English?

Reviewer #1: Yes

Reviewer #2: Yes

5. Review Comments to the Author

Reviewer #1: The manuscript by Jongen et al. deals with the important topic of high prevalence and incidence rates of STIs among young women in South Africa. Syndromic management remains the main approach to STI care in South Africa, but studies like this highlight the urgent need to move to a diagnostic care approach. The authors describe a high burden of STIs in this cohort, and describe risk factors for incident infections. These results for the incidence analysis need to be interpreted with caution as no test of cure was conducted as part of the study to definitely exclude persistent infection. Overall the manuscript adds to the growing literature of STIs in South Africa, but I suggest that the authors expand the methods section to provide more clarity on potential bias in data collection and interpretation.

Major Comments

1. The potential bias with calculating a STI incidence rate have not been adequately addressed in the manuscript. This is notoriously difficult to measure, because it is often difficult to establish whether treatment was actually taken (here it appears to be self-reported), and STIs are often persistent rather than recurrent (due to lack of treatment or treatment resistant STIs). For example, a main finding in this study is that baseline CT infection predicts incident CT. However, some of these infections may not have been treated adequately at baseline (and the study did not include a test of cure).

2. The process of providing results to participants is poorly described. How long after the test were participants informed. How many were reached, and how many accepted/ refused treatment? Furthermore, was treatment offered to partners, or contact cards distributed?

3. The issue of using reusable urine cups in 102 participants, and then retesting those participants after 2 months is not clear at all. The authors state that the results were the same, but how is this possible? Were these participants not treated at baseline? Or did they start the analysis at 2-month instead of baseline for these participants? This is a potential bias in the results and is only superficially described in the methods section. In the results section (NG section) the authors then provide a p-value for the comparison, but the number and effect size is not clear.

4. How was the questionnaire administered and demographic and behavioural data gathered? What were the potential bias with the administration of the questionnaire? Please add this to the methods section. It is briefly mentioned in the limitation section of the discussion, but not in the methods section.

5. There is very little information about partners, partner notification services and partner therapy uptake. Also, the conclusion should probably highlight this as a gap in the STI care cascade.

6. Supplementary Figure 1, the flow diagram, is key to understanding the study, and should be included in the main text of the study instead of supplementary materials.

Minor Comments

Abstract

l.43 suggest to add laboratory assay name to abstract

l.58 suggest say ‘the majority of’ rather than ‘all’ women. For example, some may not have a partner or have no sex.

Introduction

l.72 suggest change organism names to italics.

Methods

l.111 – 145 expand on how reusable cups were cleaned, and how the authors went about evaluating the validity of these results. How exactly did the comparison with the 2-month visit samples take place, and what were the findings?

l.116 How many participants refused treatment?

l.133 Where was the Anyplex stationed? In a central laboratory? What were the results turnaround times?

Results

The questionnaire results should be presented with more caution. For example, line 168, suggest to add ‘reported’ to receive education…

l.191 Not clear whether these women were not treated at baseline (see major comment).

l.229 The authors present a p-value, saying that there was no difference between baseline and 2-months. There should be an effect size, and further explanation to clarify this to the reader. It almost seems like the authors are confused by the retesting of the 102 participants themselves.

Discussion

l.260 As mentioned above, several of these infections may not have been cleared in the first place.

l.266 Suggest adding the need for better partner services and screening.

L.275 I agree with the authors that the overwhelming risk factor in this setting is young age.

l.284 ‘Tablet based questionnaires’ – this should be added to the methods as part of the expansion on the information on the questionnaire implementation.

l.288 ‘treatment failure’ or person not taking the treatment in the first place?

Conclusion

L.293 suggest adding ‘the majority’ instead of ‘all’

Reviewer #2: The paper has some interesting points. Can you please include a definition for incident STI infections. There was no mention of partner testing so it is unclear if these were re-infections or new infections. There is no data on previous pregnancies

6. PLOS authors have the option to publish the peer review history of their article (what does this mean?). If published, this will include your full peer review and any attached files.

Reviewer #1: No

Reviewer #2: No

---

## [Author Response · Author response to Decision Letter 0]

8 Jan 2021

Reviewer #1: The manuscript by Jongen et al. deals with the important topic of high prevalence and incidence rates of STIs among young women in South Africa. Syndromic management remains the main approach to STI care in South Africa, but studies like this highlight the urgent need to move to a diagnostic care approach. The authors describe a high burden of STIs in this cohort, and describe risk factors for incident infections. These results for the incidence analysis need to be interpreted with caution as no test of cure was conducted as part of the study to definitely exclude persistent infection. Overall the manuscript adds to the growing literature of STIs in South Africa, but I suggest that the authors expand the methods section to provide more clarity on potential bias in data collection and interpretation.

Response: Thank you for taking the time to review our manuscript. We appreciate the elaborate feedback.

Major Comments

Comment 1:

1. The potential bias with calculating a STI incidence rate have not been adequately addressed in the manuscript. This is notoriously difficult to measure, because it is often difficult to establish whether treatment was actually taken (here it appears to be self-reported), and STIs are often persistent rather than recurrent (due to lack of treatment or treatment resistant STIs). For example, a main finding in this study is that baseline CT infection predicts incident CT. However, some of these infections may not have been treated adequately at baseline (and the study did not include a test of cure).

Response: Thank you. Indeed, no test of cure was performed in our study and it therefore possible that infections that were now considered “incident” were actually persistent, as we also state in the Discussion. 

Original Revised Section Page (lines)

Last, although adequate treatment was offered to participants with a CT or NG infection, no test of cure was performed. Therefore, it may be possible that some of the incident cases were the result. Last, although adequate treatment was offered to participants with a CT or NG infection, no test of cure was performed. Therefore, it may be possible that some of the incident cases were the result of treatment failure or because the participant did not take the treatment. Discussion 22 (321-324)

To exclude a possible effect of persistent infections on the incidence rate, we now included three sensitivity analyses in which we excluded all women with a prevalent CT, NG, or CT and/or NG infection, respectively. The incidence rate of CT was lower when excluding participants with a prevalent CT infection at baseline, although still high (29.0 per 100 person-years). The incidence rates of NG and CT and/or NG did not differ much after exclusion of participants with a prevalent infection. The fact that the CT incidence rate was lower in this sensitivity analysis leaves open the question whether this was because treatment failures or cases not being treated were now excluded, or because re-infection was more common among women who previously had CT. 

We have now added an explanation and the results of the sensitivity analysis in the manuscript:

Original Revised Section Page (lines)

- In a sensitivity analysis, we assessed the incidence rates per 100 person-years of CT, NG, and CT and/or NG, and the determinants of these incident infections, among participants without a prevalent CT, NG, or CT and NG infection, respectively. Statistical analysis, Methods 9 (168-170)

- In sensitivity analysis among participants without a prevalent CT infection at baseline, the CT incidence rate was lower (29.0 per 100 person-years, 95% CI 20.0-42.0) and multivariable analysis did not identify determinants of an incident CT infection. Chlamydia, Results 13 (216-219)

- In sensitivity analysis, NG incidence rate was 20.2 per 100 person-years (95% CI 13.9-29.5) and multivariable analysis yielded similar results (data not shown). Gonorrhoea, Results 20 (266-267)

- In sensitivity analysis, the incidence rate of CT and/or NG infections was lower (36.3 per 100-person-years, 95% CI 26.7-49.3) and having 3-4 lifetime sex partners was no longer significantly associated with an increased risk of an incident infection. Chlamydia and/or gonorrhea, Results 20-21 (278-281)

- In sensitivity analyses among participants without prevalent infections at baseline, we showed that the incidence rate of CT, NG and CT and/or NG was still very high, although the CT incidence rate was somewhat lower than in the main analysis. The fact that the CT incidence rate was lower may be because participants with treatment failure or who did not take their treatment were now excluded, or because re-infection was more common among women with a previous CT infection. Discussion 22-23 (324-329)

Comment 2:

2. The process of providing results to participants is poorly described. How long after the test were participants informed. How many were reached, and how many accepted/ refused treatment? Furthermore, was treatment offered to partners, or contact cards distributed?

Response: The results for Chlamydia and NG were reported in batches to the study team. In cases with positive test results, participants were informed to attend the clinic at their earliest convenience for a script for syndromic treatment. This was dispensed by the local government clinic where the study office was situated. Most cases received a script for treatment but compliance with or acceptance of treatment was not captured. 

Original Revised Section Page (lines)

Participants who tested positive for HIV, CT or NG during follow-up were contacted and provided with STI results and offered treatment. Participants who tested positive for HIV, CT or NG during follow-up were contacted, provided with STI results and offered to collect treatment at their earliest convenience. Treatment was dispensed by the local government clinic where also the study office was located. Acceptance and compliance to treatment was not captured. Sexual partners of the participant were not tested for STIs or notified of exposure to STIs during the course of this study. Study procedure, Methods 6-7 (112-117)

Comment 3:

3. The issue of using reusable urine cups in 102 participants, and then retesting those participants after 2 months is not clear at all. The authors state that the results were the same, but how is this possible? Were these participants not treated at baseline? Or did they start the analysis at 2-month instead of baseline for these participants? This is a potential bias in the results and is only superficially described in the methods section. In the results section (NG section) the authors then provide a p-value for the comparison, but the number and effect size is not clear.

Response: Reusable urine cups were used for the first 102 participants, which were thoroughly cleaned after every use. As explained in the Methods section, we saw a very high prevalence of CT and NG at baseline (as reported in our baseline paper). Even though cups were cleaned according to the study protocol, we wanted to eliminate the possibility of cross-contamination as a reason for this high prevalence. In order to do so, we collected additional urine samples for the first 102 participants (of whom participants diagnosed with CT and/or NG had not been treated in-between visits and were provided with treatment at this second visit) in disposable cups at the second HPV vaccination visit. The prevalence of CT and NG at the first visit (when reusable cups were used) was comparable to the prevalence of CT and NG at the second visit (at which disposable cups were used), suggesting cross-contamination was not responsible for the initially observed high prevalence. As we found no evidence for cross-contamination, we started time at risk for the first 102 participants from two weeks after receiving treatment, as we did for the other participants. We have now added more information on the retesting of the first 102 participants, please see below. 

Original Revised Section Page (lines)

After a very high CT and NG prevalence was found at baseline [27] additional urine samples were collected in single use, disposable cups at month 2 for the first 102 participants in order to rule out cross contamination. After a very high CT and NG prevalence was found at baseline [27] additional urine samples were collected in single use, disposable cups at month 2 (during which those diagnosed with CT and/or NG picked up of treatment for their infection) for the first 102 participants in order to rule out cross contamination. Study procedures, Methods 6 (109-112)

Prevalence at baseline and month 2 for the first 102 participants was compared using Pearson’s chi square test. 

 In order to rule out possible cross-contamination due to the use of re-usable cups (although thoroughly cleaned), we compared the prevalence at baseline and month 2 for the first 102 participants using Pearson’s chi square test. If no evidence for cross-contamination was found, all these participants were included from their first study visit. If there was evidence for cross-contamination, all these participants would only be included from their second study visit (when the disposable cups were used). Statistical analysis, Methods 8 (140-146)

CT prevalence did not differ between enrolment and month 2 among women who were screened at both study visits (p=0.59, data not shown); all visits of these women were thus included in the incidence analysis. CT prevalence did not differ between enrolment (33.3%) and month 2 (29.2%) among women who were screened at both study visits (p=0.59); therefore all visits of these women were thus included in the incidence analysis. Chlamydia, Results 13 (209-212)

NG prevalence did not differ between enrolment and month 2 among women who were screened for NG at those study visits (p=0.41, data not shown); all visits of these women were included in the incidence analysis. NG prevalence did not differ significantly between enrolment (8.3%) and month 2 (12.5%) among women who were screened for NG at both these study visits (p=0.41); therefore all visits of these women were included in the incidence analysis. Gonorrhoea, Resuls 20 (258-261)

Comment 4:

4. How was the questionnaire administered and demographic and behavioural data gathered? What were the potential bias with the administration of the questionnaire? Please add this to the methods section. It is briefly mentioned in the limitation section of the discussion, but not in the methods section.

Response: All questionnaires were tablet-based and offered in three languages (English, Xhosa and Afrikaans). Questionnaires were computer-assisted and were completed by the respondents themselves without additional help. We elaborated on this in the Methods, Study procedures:

Original Revised Section Page (lines)

At enrolment and the 7-month visit, participants completed a tablet-based questionnaire on sexual history, and socio-demographic characteristics. The questionnaire was a computer-assisted self-interview and was available in English, Xhosa, and Afrikaans. At enrolment and the 7-month visit, participants completed a tablet-based questionnaire on sexual history, and socio-demographic characteristics (S1, S2, S3, S4, S5, and S6 File). To minimize social desirability bias, the questionnaire was a computer-assisted self-interview (instead of interviewer-administered) and was available in English, Xhosa, and Afrikaans. Study procedure, Methods 7 (123-127)

Comment 5:

5. There is very little information about partners, partner notification services and partner therapy uptake. Also, the conclusion should probably highlight this as a gap in the STI care cascade.

Response: During the course of our study we did not test nor notify the sexual partners of our participants. We agree that this is an important gap in current STI care and we have elaborated on this further in the Conclusion. We have now made the following changes to the Methods and Conclusion:

Original Revised Section Page (lines)

- Sexual partners of the participant were not tested for STIs or notified of exposure to STIs during the course of this study. Study procedure, Methods 7 (116-117)

Moreover, they indicate a need for screening and treatment programs to increase sexual health in this region. 

 Moreover, given the high STI prevalence in the general population of South Africa, they indicate a need for gender neutral screening and treatment programs to increase sexual health in this region. Conclusion 23 (336-338)

Comment 6:

6. Supplementary Figure 1, the flow diagram, is key to understanding the study, and should be included in the main text of the study instead of supplementary materials.

Response: As per the reviewers’ suggestion, we have now included the flow diagram in the main manuscript as Fig 1.

Minor Comments

Abstract

Comment 7:

l.43 suggest to add laboratory assay name to abstract

Response: We have now added this information to the Method section of the Abstract.

Original Revised Section Page (lines)

At enrolment and month 6 participants were screened for CT and NG. At enrolment and month 6 participants were screened for CT and NG (Anyplex CT/NG real-time detection method). Abstract 3 (38-39)

Comment 8:

l.58 suggest say ‘the majority of’ rather than ‘all’ women. For example, some may not have a partner or have no sex.

Response: We agree that ‘the majority of women’ is more suited here. We made the following changes to the Conclusion of the Abstract:

Original Revised Section Page (lines)

Besides being previously infected and a higher lifetime number of sex partners, no other risk factors were found for CT and NG, suggesting that all these women were at risk. Besides being previously infected and a higher lifetime number of sex partners, no other risk factors were found for CT and NG, suggesting that the majority of these women were at risk. Abstract 3 (52-54)

Introduction

Comment 9:

l.72 suggest change organism names to italics.

Response: Thank you, we have now used italics for the name of C. Trachomatis and N. gonorrhoeae throughout the manuscript.

Methods

Comment 10:

l.111 – 145 expand on how reusable cups were cleaned, and how the authors went about evaluating the validity of these results. How exactly did the comparison with the 2-month visit samples take place, and what were the findings?

Response: The reusable cups were thoroughly cleaned using cidex immersion for at least 15 minutes and then rinsed with tap water. We elaborated on the comparison between the prevalence results under comment 3.

Original Revised Section Page (lines)

Urine cups were thoroughly cleaned after every use. Urine cups were thoroughly cleaned after every use using cidex immersion for at least 15 minutes, after which cups were rinsed with tap water. Study procedures, Methods 6 (107-108)

Comment 11:

l.116 How many participants refused treatment?

Response: This data is not available. Participants were given a script for the clinic pharmacy, but the number of scripts dispensed and compliance with treatment was not captured. 

Comment 12:

l.133 Where was the Anyplex stationed? In a central laboratory? What were the results turnaround times?

Response: In a central laboratory at an academic hospital in the same district (Tygerberg). The turnaround time was variable due to batching of samples but was generally not more than 8 weeks. 

Original Revised Section Page (lines)

- All laboratory analyses were performed in a central laboratory at the Tygerberg hospital (Cape Town, South Africa). Laboratory analyses, Methods 7 (135-136)

Comment 13:

Results

The questionnaire results should be presented with more caution. For example, line 168, suggest to add ‘reported’ to receive education…

Response: As the questionnaires were based on self-report, we acknowledge that the presentation of the results of the questionnaire should reflect this. We have made changed accordingly to the Result section.

Original Revised Page (lines)

The majority of women included at baseline received education up to grade 12 or less (n=212, 60.4%), were single (n=334, 95.2%), and did not use birth control (n=184, 52.4%). Median age of sexual debut was 17 years [IQR 16-18] and median number of lifetime sexual partners was 2 [IQR 1-3]. Median age at baseline was 20 years [interquartile range (IQR) 19-22] (Table 1). The majority of women included at baseline reported that they had received education up to grade 12 or less (n=212, 60.4%), were single (n=334, 95.2%), and did not use birth control (n=184, 52.4%). Median reported age of sexual debut was 17 years [IQR 16-18] and median reported number of lifetime sexual partners was 2 [IQR 1-3]. Thirteen women (3.7%) at baseline indicated that they had ever received presents, money or drugs in exchange for sex. Results 9 (180-185)

Comment 14:

l.191 Not clear whether these women were not treated at baseline (see major comment).

Response: Participants collected their treatment at the month 2 visit and where thus not treated in between visits. Please see our response to comment 3. 

Comment 15:

l.229 The authors present a p-value, saying that there was no difference between baseline and 2-months. There should be an effect size, and further explanation to clarify this to the reader. It almost seems like the authors are confused by the retesting of the 102 participants themselves.

Response: We have now added the prevalence at enrolment and month 2 for the participants who were first tested with a reusable cup. Please see our response to comment 3. 

Discussion

Comment 16:

l.260 As mentioned above, several of these infections may not have been cleared in the first place.

Response: Please see our response to comment 1. 

Comment 17:

l.266 Suggest adding the need for better partner services and screening.

Response: Thank you. We agree that partner services and gender neutral screening is very important, especially when the STI prevalence is high in the general population. We already recommended this in the Discussion to include both genders in STI screening programs. We now further elaborated on this in the Conclusion.

Original Revised Section Page (lines)

As the STI prevalence in the general population of South Africa is high [7] and in the analysis we only included participants who received adequate treatment for their CT infection, it appears that repeated CT infections were common. Given that, even with a median of two lifetime sex partners, these women were at substantial risk for incident and repeated STI infections, suggesting the need to include both genders in STI screening programs in an effort to decrease STI risk. Discussion 21 (296-302)

Moreover, they indicate a need for screening and treatment programs to increase sexual health in this region. 

 Moreover, given the high STI prevalence in the general population of South Africa, they indicate a need for gender neutral screening and treatment programs to increase sexual health in this region. 

 Conclusion 23 (336-338)

Comment 18:

L.275 I agree with the authors that the overwhelming risk factor in this setting is young age.

Response: Thank you.

Comment 19:

l.284 ‘Tablet based questionnaires’ – this should be added to the methods as part of the expansion on the information on the questionnaire implementation.

Response: The questionnaires being tablet based was already previously stated in the Methods section. Please also see our response to comment 4. 

Comment 20:

l.288 ‘treatment failure’ or person not taking the treatment in the first place?

Response: Thank you, we agree that indeed incident cases could have been the results of both treatment failure and not taking the treatment. We have now added this to the Discussion.

Original Revised Page (lines)

Therefore, it may be possible that some of the incident cases were the result of treatment failure. Therefore, it may be possible that some of the incident cases were the result of treatment failure or because the participant did not take the treatment Discussion 22 (322-324)

Conclusion

Comment 21:

L.293 suggest adding ‘the majority’ instead of ‘all’

Response: We revised this accordingly in both the Discussion and the Conclusion.

Original Revised Section Page (lines)

Having a prevalent CT infection at baseline and more than three lifetime sex partners were the only determinants associated with an increased risk of CT or NG, suggesting that all these young women were at high risk for an incident infection. Having a prevalent CT infection at baseline and more than three lifetime sex partners were the only determinants associated with an increased risk of CT or NG, suggesting that the majority of these young women were at high risk for an incident infection. Discussion 21 (288-291)

Besides being previously infected with CT and a higher lifetime number of sex partners, no other risk factors were found for CT and NG, suggesting that all these women were at high risk for both STIs. Besides being previously infected with CT and a higher lifetime number of sex partners, no other risk factors were found for CT and NG, suggesting that the majority of these women were at high risk for both STIs. Conclusion 23 (332-335)

Reviewer #2: 

Comment 1: 

The paper has some interesting points. 

Response: We are very grateful to the Reviewer for taking the time to read our manuscript and we thank the Reviewer for the compliment. 

Comment 2:

Can you please include a definition for incident STI infections. 

Response: We have now included a definition for an incident CT or NG infection in the Statistical Analysis of the Methods.

Original Revised Section Page (lines)

- We defined an incident infection as the first positive CT and/or NG diagnosis after a negative CT and/or NG diagnoses or collection of adequate treatment. Statistical analysis, Methods 8 (157-159)

Comment 3:

There was no mention of partner testing so it is unclear if these were re-infections or new infections. 

Response: In this study, we were unfortunately only able to test our participants and not their partners. With that in mind, we could indeed not distinguish between re-infections and new infections. However, participants did report a median lifetime number of sexual partners of 2, while STI incidence was very high. It is thus likely that some of these incident events were re-infections, especially given that the STI prevalence in the general population of South Africa is high. We also stated this in the Discussion and this is why we advocate for gender neutral STI screening programs. Although we agree that it would have been very interesting to assess the rate of re-infection within this population as an additional research question, this would not affect the number of incident events as a re-infection is still a new infection regardless of sexual partner. 

We now included a statement in the Methods section that sexual partners were not tested for STIs nor notified for potential exposure during the course of our study. 

Original Revised Section Page (lines)

- Sexual partners of the participant were not tested for STIs or notified of exposure to STIs during the course of this study. Study procedure, Methods 7 (116-117)

Although only few lifetime sex partners (median of 2) were reported, STI incidence was high. In addition, a prevalent CT infection at baseline increased the risk 5-fold for an incident CT infection, similar to what was observed in another South African study [23]. As the STI prevalence in the general population of South Africa is high [7] and in the analysis we only included participants who received adequate treatment for their CT infection, it appears that repeated CT infections were common. Given that, even with a median of two lifetime sex partners, these women were at substantial risk for incident and repeated STI infections, suggesting the need to include both genders in STI screening programs in an effort to decrease STI risk. - Discussion 21 (294-302)

Comment 4:

There is no data on previous pregnancies

Response: In Table 1 we state the number of women who reported a previous pregnancy (i.e. 173 women reported that they had ever been pregnant).

---

## [Decision Letter · Decision Letter 1]

26 Feb 2021

PONE-D-20-31236R1

Incidence and risk factors of C. trachomatis and N. gonorrhoeae among young women from the Western Cape, South Africa: the EVRI study

PLOS ONE

Dear Dr. Jongen,

Thank you for submitting your manuscript to PLOS ONE. After careful consideration, we feel that it has merit but does not fully meet PLOS ONE’s publication criteria as it currently stands. Therefore, we invite you to submit a revised version of the manuscript that addresses the points raised during the review process.

We look forward to receiving your revised manuscript.

Kind regards,

Remco PH Peters, MD, PhD, DLSHTM

Academic Editor

PLOS ONE

Journal Requirements:

Reviewers' comments:

Reviewer's Responses to Questions

**Comments to the Author**

1. If the authors have adequately addressed your comments raised in a previous round of review and you feel that this manuscript is now acceptable for publication, you may indicate that here to bypass the “Comments to the Author” section, enter your conflict of interest statement in the “Confidential to Editor” section, and submit your "Accept" recommendation.

Reviewer #3: (No Response)

2. Is the manuscript technically sound, and do the data support the conclusions?

Reviewer #3: Partly

3. Has the statistical analysis been performed appropriately and rigorously? 

Reviewer #3: No

4. Have the authors made all data underlying the findings in their manuscript fully available?

Reviewer #3: No

5. Is the manuscript presented in an intelligible fashion and written in standard English?

Reviewer #3: Yes

6. Review Comments to the Author

Reviewer #3: I thought this was an interesting analysis that is worthwhile. As the authors note, these data arise from a trial and it's hard to grasp whether these data can be representative of a wider population. Usually, the answer with trials is no. I feel the authors used appropriate language and noted this in the limitations.

I reviewed the design and methods. Overall, this looked good, but I had a few comments below for the authors with the aim of trying to improve the statistical methods.

1. (lines 147-148) I didn't completely understand why participants lost to follow up were excluded. If you are doing a person-time calculation, then you could include everyone for the amount of time for which data are available.

2. (lines 149-151) With the uncertainty of the infection time, an interval censored survival analysis could be implemented. That type of model is specifically designed for such situations and could potentially better handle the lost to follow up.

3. (lines 162-164) If you don't utilize my prior comment, my understanding is that you've modeled the rates of events per 100 person-years. I suggest trying to model the number of incident infections instead of the rates with the person years as an offset. This is a more natural way to model this since the rates can sometimes follow some very strange distributions and have odd outliers. I think you can stick with the Poisson distribution if you like because I'm guessing the outcome will still be zeros and ones. Often, because the mean and variance are the same parameter in Poisson, that can be a very poor assumption in regression models and switching to quasipossion or negative binomial model is a better option. Though, I don't think you need that here, though I would also try to include a robust variance estimate to make sure the confidence intervals are appropriate

4. (lines 164-167) I have a few comments about the variable selection procedures utilized here. Were those variables with p < 0.20 in unvariable models the ones that are forced into the model? Otherwise, it's unclear why both bivariate screening (lines 164-5) and backward selection (lines 166-7) are used. Though, the larger problem is, with the sample sizes you have, stepwise variable selection procedures (backward included) usually do not do a good job of finding the most appropriate model (e.g., https://doi.org/10.1002/sim.3943). Stepwise procedures and any p-value based selection have quite a bit of evidence suggesting that they are poor at selecting the appropriate variables. For a decent summary, see the link above. Generally, it's better to select based on more robust criteria, especially measures which assess the fit of the model, such as BIC, or, better yet, a shrinkage-based estimator such as lasso or lars.

5. (Table 1) Given the sporadic missing data, did you consider using multiple imputation or some other method to try to handle the missing data?

6. (line 267) Since there is no word count and electronic supplements are acceptable, there is really no reason for analyses to not be reported that are important enough to be mentioned in the manuscript. Please include these analyses somewhere.

7. PLOS authors have the option to publish the peer review history of their article (what does this mean?). If published, this will include your full peer review and any attached files.

Reviewer #3: No

---

## [Author Response · Author response to Decision Letter 1]

1 Apr 2021

Reviewers' comments:

Comment 1:

I thought this was an interesting analysis that is worthwhile. As the authors note, these data arise from a trial and it's hard to grasp whether these data can be representative of a wider population. Usually, the answer with trials is no. I feel the authors used appropriate language and noted this in the limitations.

I reviewed the design and methods. Overall, this looked good, but I had a few comments below for the authors with the aim of trying to improve the statistical methods.

Response: Thank you for taking the time to review the design and methods of this study and for the kind words. 

Comment 2:

(lines 147-148) I didn't completely understand why participants lost to follow up were excluded. If you are doing a person-time calculation, then you could include everyone for the amount of time for which data are available.

Response: Testing for chlamydia and gonorrhea was done at enrolment and 6 months after the enrolment visit (line 105-106). The first 102 participants were also tested for chlamydia and gonorrhea at the month 2 visit to rule out cross contamination (line 109-112). Thus, the majority of participants had only 2 possible options for chlamydia and gonorrhea testing (enrolment and month 6). We agree that by doing a person-time calculation you can usually include data until the participant becomes lost-to-follow-up. However, if a participant missed the month 6 follow-up visit, no person-time could be included as there were only two visits. 

Comment 3:

(lines 149-151) With the uncertainty of the infection time, an interval censored survival analysis could be implemented. That type of model is specifically designed for such situations and could potentially better handle the lost to follow up.

Response: Thank you for this suggestion and please also see our response to comment 2. As we had a short follow-up (6 months) and only limited visits (2 for the majority of participants), we could not take person-time into account for women who were lost-to-follow-up. Given the short interval time between STI tests, we feel the midpoint assumption gives a fairly unbiased estimation of the timing of the incident events.

Comment 4:

(lines 162-164) If you don't utilize my prior comment, my understanding is that you've modeled the rates of events per 100 person-years. I suggest trying to model the number of incident infections instead of the rates with the person years as an offset. This is a more natural way to model this since the rates can sometimes follow some very strange distributions and have odd outliers. I think you can stick with the Poisson distribution if you like because I'm guessing the outcome will still be zeros and ones. Often, because the mean and variance are the same parameter in Poisson, that can be a very poor assumption in regression models and switching to quasipossion or negative binomial model is a better option. Though, I don't think you need that here, though I would also try to include a robust variance estimate to make sure the confidence intervals are appropriate

Response: We indeed modelled the number of incident events with the person-time as offset in a Poisson model, not the incidence rates. We made this more clear now in the Statistical Analysis.

Original Revised Section Page (lines)

Incidence rate ratios (IRR) and 95% confidence intervals (CI) of determinants of incident CT, NG and CT and/or NG infections were assessed using a Poisson regression model. Incidence rate ratios (IRR) and 95% confidence intervals (CI) of determinants of incident CT, NG and CT and/or NG infections were assessed using a Poisson regression model, using the number of incident infections as the outcome. Methods 8-9 (162-164)

Comment 5:

(lines 164-167) I have a few comments about the variable selection procedures utilized here. Were those variables with p < 0.20 in unvariable models the ones that are forced into the model? Otherwise, it's unclear why both bivariate screening (lines 164-5) and backward selection (lines 166-7) are used. Though, the larger problem is, with the sample sizes you have, stepwise variable selection procedures (backward included) usually do not do a good job of finding the most appropriate model (e.g., https://doi.org/10.1002/sim.3943). Stepwise procedures and any p-value based selection have quite a bit of evidence suggesting that they are poor at selecting the appropriate variables. For a decent summary, see the link above. Generally, it's better to select based on more robust criteria, especially measures which assess the fit of the model, such as BIC, or, better yet, a shrinkage-based estimator such as lasso or lars.

Response: Our approach was as follows: we forced age at enrolment, age at first sex act and lifetime number of sex partners into the multivariable model, based on their previously found association with STI incidence. Other variables were evaluated in univariable analysis and those associated at p<0.20 were also included in the first multivariable model. We then tried to drop other variables, and examined whether dropping them would significantly improve the fit of the model, using the likelihood-ratio test (LRT). Variables whose exclusion led to a significantly worse fit (based on the LRT) were retained. So far, not much is known about the determinants for STI incidence within this younger population of women. We therefore chose to use this approach of forcing known variables into the model and including other variables (entry based on p of univariable analysis and exit based upon LRT), instead of making a theory based model (e.g. with a DAG). 

We acknowledge our approach was not made sufficiently clear in the Statistical analysis, and we have now made the following changes.

Original Revised Section Page (lines)

Backwards selection was performed to obtain the multivariable model with the best fit. Backwards selection, using the likelihood-ratio test, was performed to obtain the multivariable model with the best fit. Methods 9 (167-168)

Comment 6:

(Table 1) Given the sporadic missing data, did you consider using multiple imputation or some other method to try to handle the missing data?

Response: Nine incident chlamydia events and 7 incident gonorrhea events were not taken into account due to missing data . Due to this limited number of missing values, a complete case analysis seemed justified. 

Comment 7:

(line 267) Since there is no word count and electronic supplements are acceptable, there is really no reason for analyses to not be reported that are important enough to be mentioned in the manuscript. Please include these analyses somewhere.

Response: We agree. We have now included the results of the sensitivity analysis as Supplementary Table 1.

---

## [Editor Report · Decision Letter 2]

16 Apr 2021

Incidence and risk factors of C. trachomatis and N. gonorrhoeae among young women from the Western Cape, South Africa: the EVRI study

PONE-D-20-31236R2

Dear Dr. Jongen,

We’re pleased to inform you that your manuscript has been judged scientifically suitable for publication and will be formally accepted for publication once it meets all outstanding technical requirements.

Kind regards,

Remco PH Peters, MD, PhD, DLSHTM

Academic Editor

PLOS ONE
---

## [Editor Report · Acceptance letter]

22 Apr 2021

PONE-D-20-31236R2 

Incidence and risk factors of *C. trachomatis* and *N. gonorrhoeae* among young women from the Western Cape, South Africa: the EVRI study 

Dear Dr. Jongen:

I'm pleased to inform you that your manuscript has been deemed suitable for publication in PLOS ONE. Congratulations! Your manuscript is now with our production department. 

Kind regards, 

on behalf of

Prof Remco PH Peters 

Academic Editor

PLOS ONE